# Topological control of extreme waves

Giulia Marcucci [1,2]*, Davide Pierangeli[1,2], Aharon J. Agranat[3], Ray-Kuang Lee [4], Eugenio DelRe[1,2] & Claudio Conti [1,2]

From optics to hydrodynamics, shock and rogue waves are widespread. Although they appear as distinct phenomena, transitions between extreme waves are allowed. However, these have never been experimentally observed because control strategies are still missing. We introduce the new concept of topological control based on the one-to-one correspondence between the number of wave packet oscillating phases and the genus of toroidal surfaces associated with the nonlinear Schrödinger equation solutions through Riemann theta functions. We demonstrate the concept experimentally by reporting observations of supervised transitions between waves with different genera. Considering the box problem in a focusing photorefractive medium, we tailor the time-dependent nonlinearity and dispersion to explore each region in the state diagram of the nonlinear wave propagation. Our result is the first realization of topological control of nonlinear waves. This new technique casts light on shock and rogue waves generation and can be extended to other nonlinear phenomena.

[1] Department of Physics, University Sapienza, Piazzale Aldo Moro 5, 00185 Rome, Italy. [2] Institute for Complex Systems, Via dei Taurini 19, 00185 Rome, Italy. [3] Applied Physics Department, Hebrew University of Jerusalem, 91904 Jerusalem, Israel. [4] Institute of Photonics Technologies, National Tsing Hua University, Hsinchu 300, Taiwan. *email: giulia.marcucci@uniroma1.it

n 1967 Gardner, Greene, Kruskal, and Miura developed a mathematical method—the inverse scattering transform (IST)[1]—disclosing the inner features of nonlinear waves in hydrodynamics, plasma physics, nonlinear optics and many other physical systems[2–4]. According to IST, one also predicts the periodical regeneration of the initial state, as in the Fermi-Pasta-Ulam-Tsingou recurrence[5,6].

The nonlinear Schrödinger equation (NLSE)[7] is a cornerstone of IST for detailing dispersive phenomena, such as dispersive shock waves (DSWs)[8–10], rogue waves (RWs)[11–14], and shape invariant solitons[15–17]. DSWs regularize catastrophic discontinuities by means of rapid oscillations[18–22]. RWs are giant disturbances appearing and disappearing abruptly in a nearly constant background[23–34]. Solitons are particle-like dispersion-free wave packets that can form complex interacting assemblies, ranging from crystals to gases[15,16,33,35–37].

DSWs, RWs, and soliton gases (SGs) are related phenomena, and all appear in paradigmatic nonlinear evolutions, such as the box problem for the focusing NLSE[38–43]. However, for the box problem in the small-dispersion NLSE, IST becomes unfeasible. In this extreme regime, the problem can be tackled by the so-called finite-gap theory[40,44]. It turns out that extreme waves are described in terms of one single mathematical entity, the Riemann theta function, and classified by a topological index, the genus $g$ (see Fig. 1). In nonlinear wave theory, $g$ represents the number of oscillating phases and evolves during light propagation: "single phase" DSWs have $g = 1$, RWs have $g \sim 2$ and SGs have $g \gg 2$. This creates a fascinating connection between extreme waves and topology. Indeed, the same genus $g$ allows a topological classification of surfaces, to distinguish, for examples, a torus and sphere (Fig. 1). The question lies open if this elegant mathematical classification of extreme waves can inspire new applications. Can it modify the basic paradigm by which the asymptotic evolution of a wave is encoded in its initial shape, opening the way to controlling extreme waves, from lasers to earthquakes?

Here, inspired by the topological classification, we propose and demonstrate the use of topological indices to control the generation of extreme waves with varying genera $g$[41]. We consider the NLSE box problem where, according to recent theoretical results[40], light experiences various dynamic phases during propagation, distinguished by different genera. In particular, for high values of a nonlinearly scaled propagation distance $\zeta$, one has $g \sim \zeta$. By continuously varying $\zeta$, we can change $g$ and explore all the possible dynamic phases (see Fig. 1, where $\zeta$ is given in terms of the observation time $t$, detailed below). We experimentally test this approach in photorefractive materials, giving evidence of an unprecedented control of nonlinear waves, which allows the first observation of the transition from focusing DSWs to RWs.

## Results

**Time-dependent spatial box problem.** We consider the NLSE

$$\imath \epsilon \partial_\zeta \psi + \frac{\epsilon^2}{2}\partial_\xi^2 \psi + |\psi|^2 \psi = 0, \tag{1}$$

where $\psi = \psi(\xi, \zeta)$ is the normalized complex field envelope, $\zeta$ is the propagation coordinate, $\xi$ is the transverse coordinate and $\epsilon > 0$ is the dispersion parameter. We take a rectangular barrier as initial condition

$$\psi(\xi, 0) = \begin{cases} q & \text{for } |\xi| \leq l \\ 0 & \text{elsewhere} \end{cases}, \tag{2}$$

that is, a box of finite height $q > 0$, length $2l > 0$, and genus $g = 0$. In our work, we fix $q = l = 1$. Equation (1) with (2) is known as the NLSE box problem, or the dam break problem,

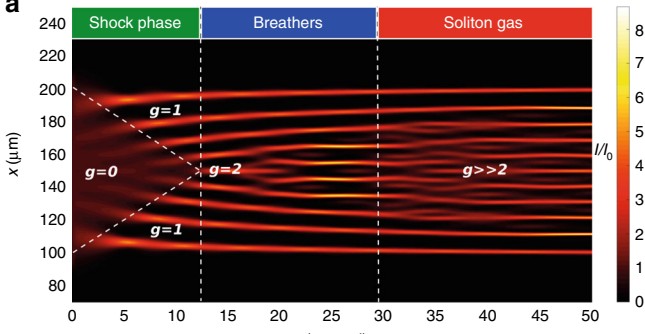

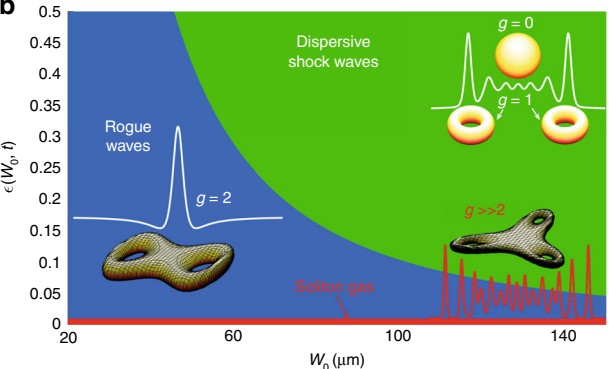

**Fig. 1** Topological classification of extreme waves. **a** Final states of the wave for a fixed initial waist $W_0 = 100\,\mu$m showing the generation of focusing dispersive shock waves ($g = 1$), rogue waves ($g \sim 2$), and a soliton gas ($g \gg 2$) after different time intervals in a photorefractive material (see text). **b** Phase diagram reporting the final states in terms of the parameter $\epsilon$ and the initial beam waist. Transitions occur by fixing waist and varying $\epsilon$ or, equivalently, the observation time $t$. Different surfaces displayed in proximity of the various wave profiles, corresponding to the different regions in the phase diagram, outline the link between the topological classification of extreme waves in terms of the genus $g$ and the topological classification of toroidal Riemann surfaces (for a sphere, $g = 0$, for a torus, $g = 1$, etc.)

which exhibits some of the most interesting dynamic phases in nonlinear wave propagation[40,45]. The initial evolution presents the formation of two wave trains counterpropagating that regularize the box discontinuities. These wave trains are single-phase DSWs ($g = 1$). Their two wavefronts superimpose in the central part of the box (see Fig. 1a)–occurring at $\zeta = \zeta_0 := \frac{l}{2\sqrt{2}q}$ – and generate a breather lattice of genus $g = 2$, a two-phase quasi-periodic wave resembling an ensemble of Akhmediev breathers (ABs)[13,28]. Since both the $\xi-$ and $\zeta-$ periods increase with $\zeta$, the oscillations at $\xi \simeq 0$ become locally approximated by Peregrine solitons (PSs)[13,46–48]. At long propagation distances $\zeta \gg \zeta_0$, the wave train becomes multi-phase and generates a SG with $g \sim \zeta$.

In Fig. 1a, we report the wave dynamics in physical units, as we make specific reference to our experimental realization of the NLSE box problem for spatial optical propagation in photorefractive media (PR). In these materials, the optical nonlinearity is due to the time-dependent accumulation of free carriers that induces a time-varying low-frequency electric field. Through the electro-optic effect, the charge accumulation results into a time-varying nonlinearity. The corresponding time-profile can be controlled by an external applied voltage and the intensity level[49–51]. These features enable to experimentally implement our topological control technique. In PR, Eq. (1) describes an optical beam with complex amplitude $A(z, x, t)$ and intensity $I = |A|^2$

through the transformation (see Methods)

$$\zeta = \frac{z}{\epsilon z_D}, \quad \xi = \frac{2x}{W_0}, \quad \psi = \frac{A}{\sqrt{I_0}}, \quad (3)$$

with $W_0$ the initial beam waist along $x$-direction, $z_D = \frac{\pi n_0 W_0^2}{2\lambda}$ the diffraction length, $n = n_0 + \frac{2\delta n_0 I}{I_S} f(t)$ the refractive index, $\delta n_0 > 0$ the nonlinear coefficient, $I_S$ the saturation intensity, $I_0$ the initial intensity. For PR

$$\epsilon = \frac{\lambda}{\pi W_0} \sqrt{\frac{I_S}{2n_0 \delta n_0 I_0 f(t)}}, \quad (4)$$

namely, the dispersion is modulated by the time-dependent crystal response function $f(t) = 1 - \exp(-t/\tau)$, with the saturation time $\tau$ fixed by the input power and the applied voltage[5].

**Genus control**. For a given propagation distance $L$ (the length of the photorefractive crystal), the genus of the final state is determined by the detection time $t$, which determines $\epsilon$, $\zeta = \frac{L}{\epsilon z_D}$, and $g$,

correspondingly. The genus time-dependence is sketched in Fig. 1a. The output wave profile depends on its genus content, which varies with $t$.

Following the theoretical approach in ref. [40], the two separatrix equations divide the evolution diagram in Fig. 1a into three different areas: the flat box plateau with genus $g = 0$, the lateral counterpropagating DSWs with genus $g = 1$, and the RWs after the DSW-collision point (corresponding to the separatrices intersection) with genus $g = 2$. The two separatrices (dashed lines in Fig. 1a) have equations

$$x = x_0 \pm \frac{W_0}{2t_0}(t - t_0) = x_0 \pm v(t - t_0), \quad (5)$$

with $(t_0, x_0)$ the DSW-collision point, $t_0 \simeq \frac{\tau I_S n_0 W_0^2}{64 I_0 \delta n_0 L^2}$, and $x_0$ given by the central position of the box. It turns out that the shock velocity is

$$v = \frac{W_0}{2t_0} = \frac{32 \delta n_0 L^2}{I_S n_0 W_0^2 U_0 \tau} P, \quad (6)$$

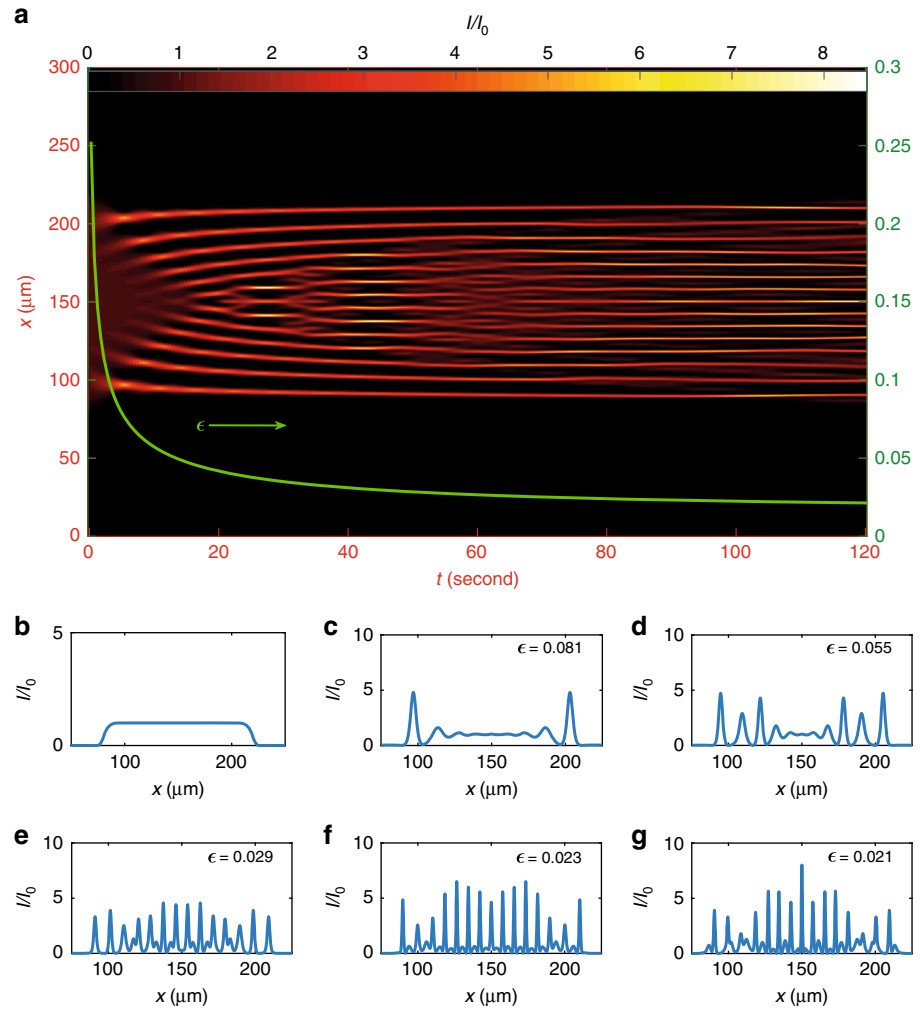

**Fig. 2** Controlling the extreme wave genus. **a** Numerical simulation of the control of the final state after a propagation distance $L = 2.5$ mm for an initial beam waist $W_0 = 140$ μm ($I_0 = \frac{P}{U_0 W_0} = 0.38 \times 10^5$ W/m²). Axis $x$ represents the beam transverse direction, axis $t$ the time of output detection. **b** Initial beam intensity: a super-Gaussian wave centered at $x = 150$ μm of height $I_0$ and width $W_0$. **c, d** Focusing dispersive shock waves occurrence: **c** represents the beam intensity at $t = 5$ s, when the wave breaking has just occurred, so two lateral intense wave trains regularize the box discontinuity and start to travel towards the beam central part; **d** the beam intensity at $t = 11$ s, which exhibits the two counterpropagating DSWs reaching the center $x = 150$ μm. **e–g** Akhmediev breathers and Peregrine solitons generation: beam intensity at **e** $t = 49$ s, **f** $t = 98$ s, and **g** $t = 120$ s, after the two dispersive shock waves superposition and the formation of Akhmediev breathers with period increasing with $t$. Since a Peregrine soliton is an Akhmediev breather with an infinite period, increasing $t$ is tantamount to generating central intensity peaks, locally described by Peregrine solitons

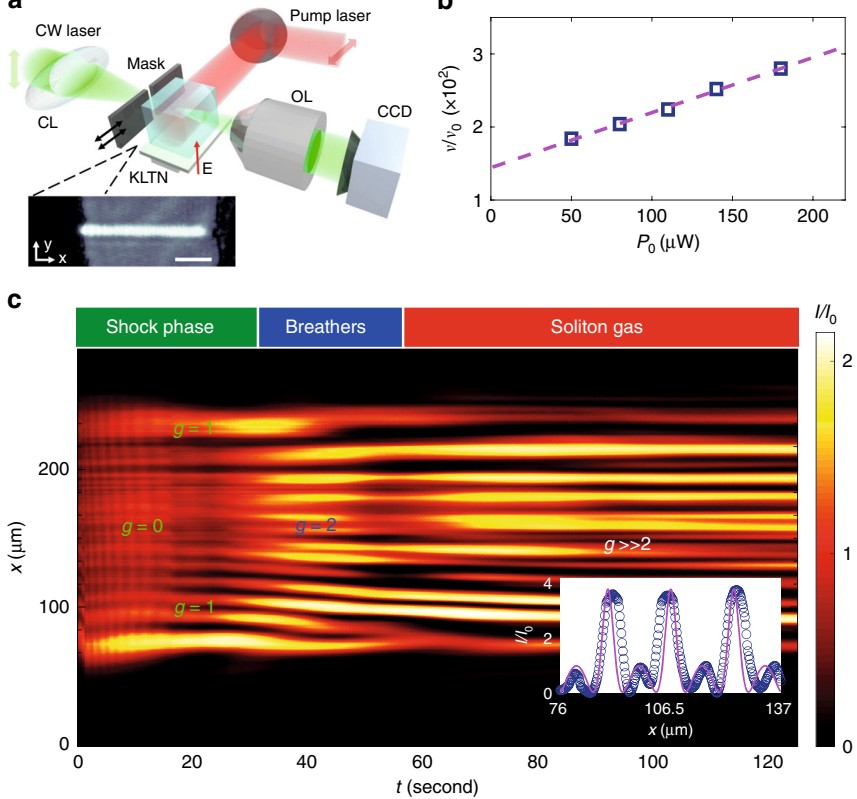

**Fig. 3** Experimental demonstration of the extreme wave genus control. **a** Experimental setup. A CW laser is made a quasi-one-dimensional wave by a cylindrical lens (CL), then a tunable mask shapes it as a box. Light propagates in a pumped photorefractive KLTN crystal, it is collected by a microscope objective and the optical intensity is detected by a CCD camera. The inset shows an example of the detected input intensity distribution (scale bar is 50 μm). **b** Normalized shock velocity [$v_0 = L/\bar{t}$, $L = 2.5$ mm, $\bar{t} = (30 \pm 2)$ s], measured through the width of the oscillation tail at fixed time, versus input power. The blue squares are the experimental data, while the dashed pink line is the linear fit. **c** Experimental observation of optical intensity $I/I_0$ for an initial beam waist $W_0 = 140$ μm. Axis $x$ represents the beam profile, transverse to propagation, collected by the CCD camera, while axis $t$ is time of CCD camera detection. Output presents a first dispersive-shock-wave phase, a transition to a phase presenting Akhmediev breather structures and, at long times, a generation of a soliton gas. The inset is an exemplary wave intensity profile detected at $t = 63$ s (dotted blue line), along with the theoretical Akhmediev breather profile

proportional to the input power, as experimentally demonstrated and detailed below.

Equation (5) expresses the genus time-dependence for its first three values $g = 0, 1, 2$. It allows designing the waveshape, before the experiment, by associating a specific combination of the topological indices, and to predict the detection time corresponding to the target topology. In other words, by properly choosing the experimental conditions, we can predict the occurrence of a given extreme wave by using the expected genus $g$. According to Eq. (4), we use time $t$ and initial waist $W_0$ to vary $\epsilon$. The accessible states are outlined in the phase diagram in Fig. 1b, in terms of $\epsilon$ and $W_0$. Choosing $W_0 = 100$ μm as in Fig. 1a, by varying $t$ one switches from DSWs to RWs, and then to SGs.

**Supervised transition from shock to rogue waves**. The case $W_0 = 140$ μm is illustrated in Fig. 2a by numerical simulations. The two focusing DSWs and the SG are visible at the beginning and at the end of temporal evolution, respectively (see phase diagram in Fig. 1b). As soon as an initial super-Gaussian wave (Fig. 2b, see Methods) starts to propagate, two DSWs appear on the beam borders (Fig. 2c) and propagate towards the beam central part (Fig. 2d). Experimental proof of the genuine non-linear nature of the beam evolution at this regime, not due to modulation instability arising from noise in the central part of the

box, is shown in Supplementary Information (Suppl. Fig. 1). When the DSWs superimpose, ABs are generated (Fig. 2e). From the analytical NLSE solutions for the focusing dam break problem[40], we see that ABs have $\xi$-period increasing with $\zeta$. Moreover, one finds that $\partial_t \zeta > 0$, therefore the period in the $x$-direction must increase with time, and central peaks appear upon evolution. These peaks are well approximated by PSs, for large $t$, as confirmed by Fig. 2f, g.

The occurrence of RWs in the large box regime is proved also by statistical analysis, illustrated in Supplementary Information (Suppl. Fig. 2h, i).

Figure 3 shows the experimental observation of the controlled dynamics simulated in Fig. 2. Figure 3a sketches the experimental setup, detailed in Methods. A quasi-one-dimensional box-shaped beam propagates in a photorefractive crystal, and the optical intensity distribution is detected at different times. The observations of shock velocities and beam propagation for $W_0 = 140$ μm are reported in Fig. 3b, c, respectively. In Fig. 3c, we see an initial DSW phase that evolves into a train of large amplitude waves. In this regime, we identify a breather-like structure (ABs, inset in Fig. 3c) that evolves into a SG at large propagation time. The DSW phase is investigated varying the input power. We find a linear increasing behavior of the shock velocity when increasing the power (Fig. 3b), as predicted by Eq. (6). The shock velocity is proportional to the distance between the two counterpropagating DSWs at a fixed time. We measured the width $\Delta x$ of the plateau

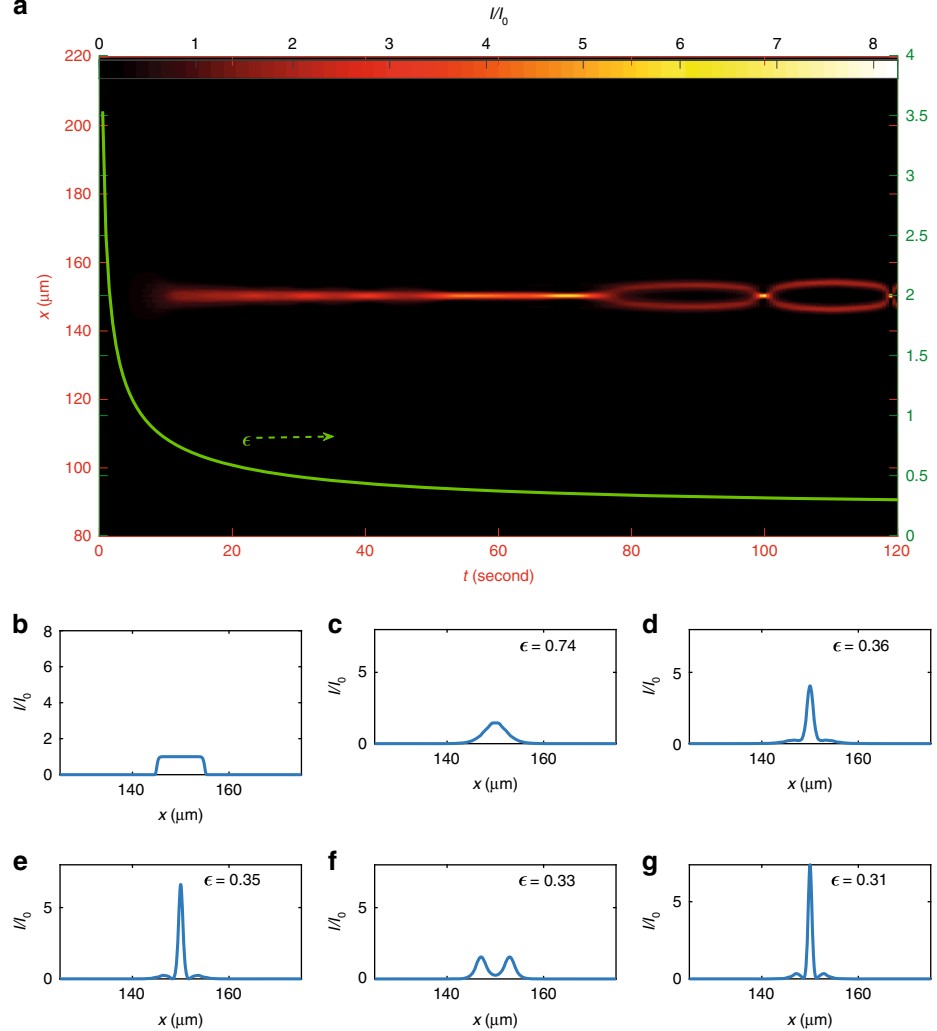

**Fig. 4** Simulation of the topological control for a small waist. **a** Numerical simulation of the control of the final state after a propagation distance $L = 2.5$ mm for an initial beam waist $W_0 = 10$ μm ($I_0 = \frac{P}{U_0 W_0} = 5.33 \times 10^5$ W/m$^2$). Axis $t$ expresses time of detection, while $x$ is the beam transverse coordinate. **b** Initial beam intensity: a super-Gaussian wave centered at $x = 150$ μm. **c–e** Peregrine soliton generation: beam intensity (**c**) at $t = 12$ s, and (**d**) at $t = 64$ s, during the formation of the Peregrine soliton, while (**e**) exhibits the Peregrine soliton profile at $t = 70$ s. **f, g** Higher-order Peregrine soliton generation: beam intensity at **f** $t = 85$ s, and **g** $t = 100$ s, where the Peregrine soliton is alternately destroyed and reformed

at time $\bar{t} \sim 30$ s. Referring to Eq. (6), we obtain the normalized velocity $\bar{v} = v/v_0$, with $v_0 = L/\bar{t}$.

**Peregrine solitons emergence.** Figure 4 illustrates the numerically determined dynamics at smaller values of the beam waist ($W_0 = 10$ μm), a regime in which the generation of single PSs is evident. The intensity profile is reported in Fig. 4a. As shown in Fig. 1b, one needs to carefully choose $W_0$ for observing a RWs generation without the DSWs occurrence. For $W_0 = 10$ μm, the super-Gaussian wave (Fig. 4b) generates a PS (Fig. 4c–e). The following dynamics shows the higher-order PS emergence (Fig. 4f, g), each order with a higher genus.

Figure 5a–g report the experimental results for the case $W_0 = 30$ μm. Observations of the Peregrine-like soliton generation are shown, both in intensity (Fig. 5a–d) and in phase (Fig. 5e–g). For a small initial waist, a localized wave, well described by the PS (Fig. 5b, d), forms and recurs without a visible wave breaking. This dynamics is in close agreement with simulations in Fig. 4d–g, where the PS is repeatedly destroyed and generated, each time at a higher order. Phase measurements are

illustrated in Fig. 5e–g. Each PS has two-phase signatures: a longitudinal smooth phase shift of $2\pi$ and a transversal rectangular phase shift profile, with height $\pi$ and basis as wide as the PS width[47,48]. Such signatures are here both experimentally demonstrated. From Fig. 5e, which shows the interference pattern during the first PS occurrence, we obtain the longitudinal phase shift behavior in Fig. 5g, by a cosinusoidal fitting along the central propagation outline. Figure 5f reports the experimental transversal phase shift profile along $x$. A comparison with the measured interference fringes is also illustrated in the inset, which directly shows the phase jump (topological defect). Stressing the significance of these results is very important, because they are a proof of the topological control: the genus is determined by the input waist and time of detection. Indeed, the longitudinal phase shift represents the transition from genus 0 to 2, whereas the transverse PS phase shift outline unveils the value $g = 2$, equal to the number of phase jumps (first from 0 to $\pi$, then again from $\pi$ to 0). This is summarized in Fig. 5h, which sketches numerical simulations of phase behavior at $W_0 = 10$ μm, normalized in $[-\pi, \pi]$. Figure 5h gives a picture of genera changes, PS occurrence and phase discontinuities. The genus is zero and the

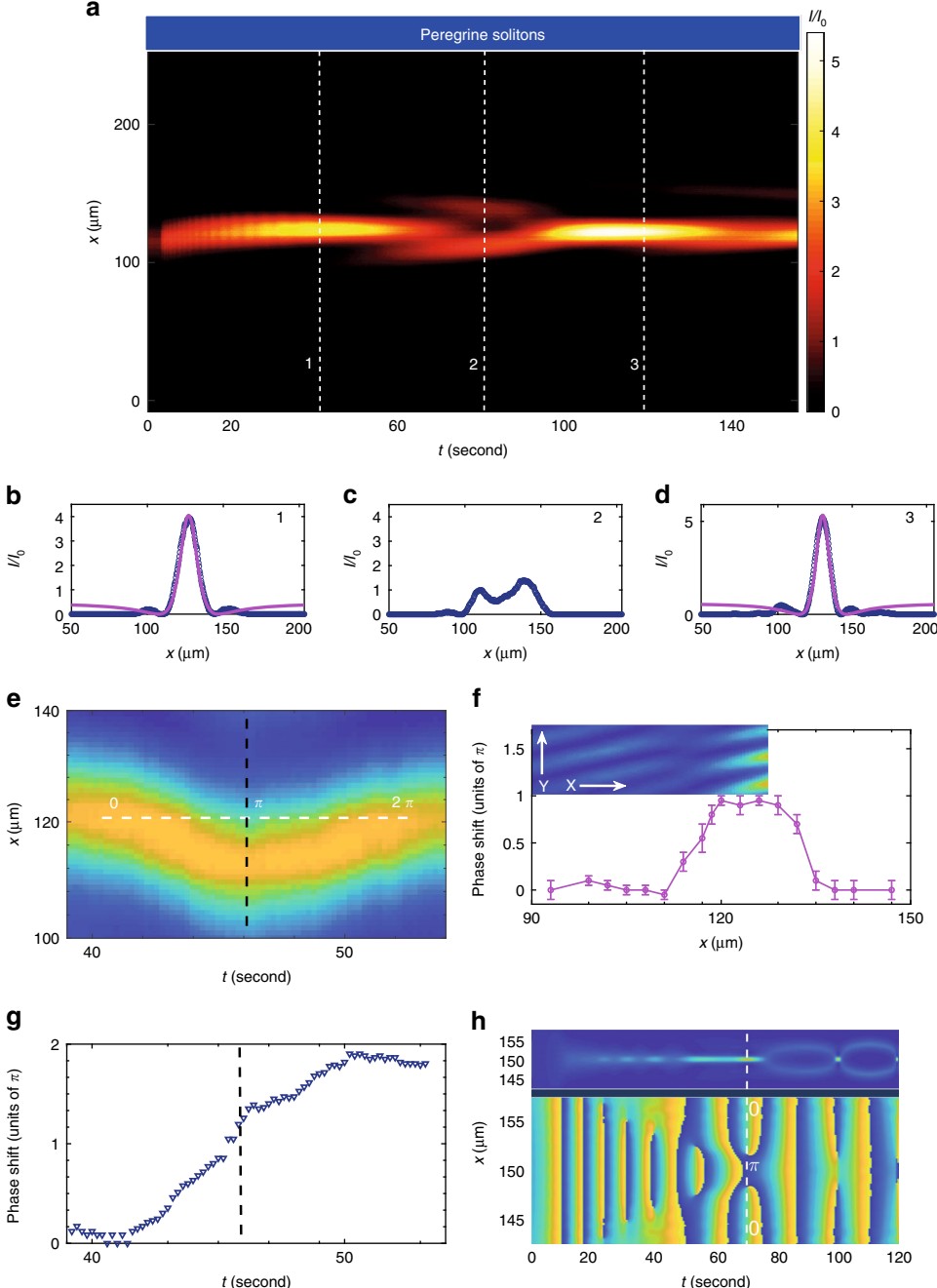

**Fig. 5** Experimental topological control for a small waist. **a** Observation of optical intensity $I/I_0$ for an initial beam waist $W_0 = 30\,\mu m$. Axis $t$ is time of output detection, $x$ is the transverse direction. In this regime, we observe Peregrine-soliton-like structures formation (see Fig. 1b) [the colored scale goes from 0 (dark blue) to 5 (bright yellow)]. **c, d** Intensity outlines corresponding to numbered dashed lines in (**a**): the blue lines are experimental waveforms, the pink continuous lines are fitting functions according to the analytical PS profile. **e–h** Phase measurements (**e–g**) and simulations (**h**) of the Peregrine soliton. The detected interference pattern during the first PS generation is reported in (**e**), corresponding to (**b**). The jump from 0 to $2\pi$ along the white dashed line corresponds to the transition from $g = 0$ to $g = 2$. The black dashed line highlights the jump, shown in (**g**). The experimental transversal phase shift profile along $x$ is reported in (**f**), showing the expected $\pi$ shift corresponding to (**b**). Error bars represent standard deviation. The inset shows the corresponding area of the measured interference fringes on the transverse plane. Phase simulations at $W_0 = 10\,\mu m$ are reported in the bottom panel in (**h**) [the colored scale goes from $-\pi$ (bright yellow) to $\pi$ (dark blue) (0 is green)]. Top panel sketches Fig. 4a, for at-a-glance correspondence between genera changes, PS occurrence and phase discontinuities

phase profile is flat until the first PS occurrences. After that, the phase value changes and the phase transverse profile presents two jumps of $\pi$.

The statistical properties of the PS intensity are illustrated in Supplementary Information (Suppl. Fig. 2f, g), and they confirm the occurrence of RWs in the small box regime.

## Discussion

The topological classification of nonlinear beam propagation by the genera of the Riemann theta functions opens a new route to control the generation of extreme waves. We demonstrated the topological control for the focusing box problem in optical propagation in photorefractive media. By using the time-dependent

photorefractive nonlinearity, we could design the final state of the wave evolution in a predetermined way and explore all the possible dynamic phases in the nonlinear propagation.

Such a novel control strategy enabled the first observation of the continuous transition from dispersive shock to rogue waves and soliton gases, demonstrating that different extreme wave phenomena are deeply linked, and also that a proper tuning of their topological content in their nonlinear evolution allows transformations from one state to another. The further numerical and experimental analysis reported in Supplementary Information proves that this new control paradigm in third-order media has a broad range of validity, where it is not affected by linear effects, like modulation instability or loss, but its nature is genuinely nonlinear.

In conclusion, our result is the first example of the topological control of integrable nonlinear waves. This new technique casts light on dispersive shock waves and rogue wave generation. It is general, not limited to the photorefractive media, and can be extended to other nonlinear phenomena, from classical to quantum ones. These outcomes are not only important for fundamental studies and control of extreme nonlinear waves, but further developments in the use of topological concepts in nonlinear physics can allow innovative applications for engineering strongly nonlinear phenomena, as in spatial beam shaping for microscopy, medicine and spectroscopy, and coherent supercontinuum light sources for telecommunication.

## Methods

**Photorefractive media.** Starting from Maxwell's equations in a medium with a third-order-nonlinear polarization, in paraxial and slowly varying envelope approximations, one can derive the propagation equation of the complex optical field envelope $A(x, y, z)$:

$$\imath \partial_z A + \frac{1}{2k} \nabla^2 A + \frac{k}{n_0} \delta n(I) A = 0, \tag{7}$$

with $z$ the longitudinal coordinate, $x, y$ the transverse coordinates and $n = n_0 + \delta n(I)$ the refractive index, weakly depending on the intensity $I = |A|^2 (\delta n(I) << n_0)$.

Equation (7) is the nonlinear Schrödinger equation (NLSE) and rules laser beam propagation in centrosymmetric Kerr media. For PR, the refractive index perturbation depends also parametrically on time, i.e., $\delta n = \delta n(I, t)$. In fact, the amplitude of the nonlinear self-interaction increases, on average, with the exposure time up to a saturation value, on a slow timescale, typically seconds for peak intensities of a few kWcm$^{-2}$ [51].

In our centrosymmetric photorefractive crystal, at first approximation $\delta n = \frac{-\delta n_0}{\left(1 + \frac{I}{I_S}\right)^2} f(t)$, with $f(t)$ the response function. $\delta n_0$ includes the electro-optic effect coefficient [49–51]. For weak intensities $I << I_S$, we obtain a Kerr-like regime with $\delta n = 2 \delta n_0 \frac{I}{I_S} f(t)$, apart from a constant term. We consider the case $\partial_y A \sim 0$ (strong beam anisotropy), thus we look for solutions of the $(1 + 1)$-dimensional NLSE for the envelope $A \sim A(x, z)$:

$$\imath \partial_z A + \frac{1}{2k} \partial_x^2 A + 2 \rho(t) |A|^2 A = 0, \tag{8}$$

with $\rho(t) = \frac{2\pi \delta n_0}{\lambda I_S} f(t)$ and the field envelope initial profile

$$A(x, 0) = \begin{cases} \sqrt{I_0} & \text{for } |x| \leq \frac{1}{2} W_0 \\ 0 & \text{elsewhere} \end{cases}. \tag{9}$$

One obtains Eq. (8) from Eq. (1) through the transformation (3). We stress that, in this case, the dispersion parameter depends on time, as follows from Eq. (4).

**Numerical simulations.** We solve numerically Eq. (1) by a one-parameter-depending beam propagation method (BPM) with a symmetrized split-step in the code core [52]. We use a high-order super-Gaussian initial condition

$$\psi(\xi, \zeta = 0) = q \exp\left\{ -\frac{1}{2} \left(\frac{\xi}{l}\right)^{24} \right\}. \tag{10}$$

For each temporal value, Eq. (1) solutions have different dispersion parameter $\epsilon$ and final value of $\zeta$, because from Eq. (3) it reads $\zeta_{\text{fin}} = \frac{4L}{\epsilon(t) k W_0^2}$, where $L$ is the crystal length. In Fig. 2 and 4, we show the numerical results. The propagation in time considers $\psi(\xi, \zeta_{\text{fin}})$, which corresponds to detections at end of the crystal.

**Experimental setup.** A $y$-polarized optical beam at wavelength $\lambda = 532$ nm from a continuous 80 mW Nd:YAG laser source is focused by a cylindrical lens down to a quasi-one-dimensional beam with waist $U_0 = 15$ µm along the $y$-direction. The initial box shape is obtained by a mask of tunable width, placed in proximity of the input face of the photorefractive crystal. A sketch of the optical system is shown in Fig. 3a. The beam is launched into an optical quality specimen of $2.1^{(x)} \times 1.9^{(y)} \times 2.5^{(z)}$ mm $K_{0.964}Li_{0.036}Ta_{0.60}Nb_{0.40}O_3$ (KLTN) with Cu and V impurities ($n_0 = 2.3$). The crystal exhibits a ferroelectric phase transition at the Curie temperature $T_C = 284$ K. Nonlinear light dynamics are studied in the paraelectric phase at $T = T_C + 8$ K, a condition ensuring a large nonlinear response and a negligible effect of small-scale disorder [53]. The time-dependent photorefractive response sets in when an external bias field $E$ is applied along $y$ (voltage $V = 500$ V). To have a so-called Kerr-like (cubic) nonlinearity from the photorefractive effect, the crystal is continuously pumped with an $x$-polarized 15 mW laser at $\lambda = 633$ nm. The pump does not interact with the principal beam propagating along the $z$ axis and only constitutes the saturation intensity $I_S$. The spatial intensity distribution is measured at the crystal output as a function of the exposure time $t$ by means of a high-resolution imaging system composed of an objective lens (NA = 0.5) and a CCD camera at 15 Hz.

In the present case, evolution is studied at a fixed value of $z$ (the crystal output) by varying the exposure time $t$. In fact, the average index change grows and saturates according to a time dependence well defined by the saturation time $\tau \sim 100$ s once the input beam intensity, applied voltage, and temperature have been fixed.

## Data availability

All data are available in this submission.

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

## Acknowledgments

We acknowledge M. Conforti, S. Gentilini, P.G. Grinevich, A. Mussot, P.M. Santini, S. Trillo, and V.E. Zakharov for fruitful conversations on related topics. We thank MD Deen Islam for technical support in the laboratory. The present research was supported by PRIN 2015 NEMO project (grant number 2015KEZNYM), H2020 QuantERA QUOMPLEX (grant number 731473), H2020 PhoQus (grant number 820392), PRIN 2017 PELM (grant number 20177PSCKT), Sapienza Ateneo (2016 and 2017 programs), and Ministry of Science and Technology of Taiwan (105-2628-M-007-003-MY4).

## Author contributions

G.M. and D.P. equally contributed to this work. G.M., R.K.L., and C.C. conceived the idea and the theoretical framework; D.P., E.D., and C.C. conceived its experimental realization. G.M. and C.C. developed the theoretical background. G.M. performed the numerical simulations. D.P. carried out experiments and data analysis. A.J.A. designed and fabricated the photorefractive crystal. All authors discussed the results and wrote the paper.

## Competing interest

The authors declare no competing interests.
