## [Peer Review File · Nature Communications]

Reviewers' Comments:

Reviewer #1:

Remarks to the Author:

Referee report for the manuscript “Topological Control of Extreme Waves” by G. Marcucci *et al.*

The manuscript reports experimental results on the evolution of an optical pulse with initial near-rectangular intensity profile in a photorefractive crystal with Kerr nonlinearity. The results are interpreted using the recent theoretical developments of Ref [31] where the “box” problem for the small-dispersion focusing nonlinear Schrödinger (NLS) equation was studied analytically and numerically. The main claim of the manuscript is that, by varying the medium’s nonlinearity the authors can control the “topology” of the nonlinear coherent structures generated in the optical pulse evolution. The topology is understood in terms of the genus of the underlying spectral Riemann surface and of the associated algebraic theta functions describing the asymptotic NLS solutions in the semi-classical limit.

The main idea of the paper looks attractive and promising, and the results are potentially significant. However, there are a number of serious issues, which should be properly addressed before the publication of the manuscript can be considered.

MAIN ISSUES

1. **The genus.** How can the authors guarantee that the structures they observe have certain genus? In other words, how much of the actual control over the wave “topology” do they have? The fact that the intensity profile they observe can be reasonably well fitted by the shape of a certain NLS breather solution (like in Fig 3 (Akhmediev) or Fig. 5 (Peregrine)) does not necessarily imply that the observed coherent structure is described by this particular NLS solution as the amplitude profile is only a “half” of the complex wave field. To be able to make the identification like this, the phase comparison is necessary along with the amplitude/intensity comparison. Can the authors compare the phase of the experimentally observed structures with that of the theoretical solution to support their claim?
2. **Modulational instability.** This issue is related to the first one. It is known that the breather structures naturally arise due to the development of modulational instability of a plane wave perturbed by a small noise (D. Agafontsev and V. Zakharov, *Nonlinearity*, 28 (2015) 2791). Thus, the coherent structures observed in the experiment reported in the manuscript could in principle arise as a result of the development of the noise induced modulational instability in the central part of the box. If the authors insist that the dynamics they observe are dominated by the “genuine” NLS box evolution, they should discuss the influence of the noise, which is inevitable in a physical experiment, and present the corresponding estimates supporting their claims. This issue is crucial, since if the effects the authors observe are dominated by modulational instability, there is no question of any “topological control”.
3. **Dissipation.** Similar to modulational instability, inevitable effects of dissipation should be discussed and estimated. In fact, the dissipation can significantly affect the

“local” genus of the breather structure, see S. Randoux et al. Phys. Rev. E 98 (2018) 022219

4. **DSW evolution and the “topology” control.** The wave topology control in the manuscript is verified against the simple formula (5) describing the motion of DSW boundaries (the so-called “breaking curves”) separating the regions with genus 0 and 1, with the point of intersection being associated with the formation of a rogue wave (genus 2). For the comparison of the theoretical velocities (6) with experiment the authors refer to Fig 3.b) showing perfect agreement but I don’t think they explain how the speeds are measured in the experiment. This is important as the breaking curves are not seen by eye on the experimental space-time diagram in Fig 3c. Furthermore, the result presented in Fig. 5 looks very rough. I do not see how it can be compared with Fig. 4 to support the topology control claim.

5. **Finite genus solutions vs. rogue waves vs. soliton gas.** The connection between rogue waves and finite gap NLS solutions has been discussed in detail in: M. Bertola, G. El and A. Tovbis, Proc. Roy. Soc., 472 (2016) 20160340. In particular, the above authors show that rogue waves can appear, under certain conditions, in the NLS solution of ANY genus greater than one, not necessarily genus 2 (so the association of rogue waves with genus 2 in the phase diagram Fig 1b is misleading). However, the present manuscript implies that any solution with $g > 2$ is not a rogue wave but a “soliton gas” (see Fig. 2). This statement is, again, misleading and, strictly speaking, incorrect. In any case, since the authors never explain what they mean by soliton gas it is difficult to meaningfully discuss this issue. At least short discussion should be present in the manuscript, in particular, because the concept of soliton gas is much less developed than that of a DSW or a rogue wave with just a handful of references following the seminal Zakharov’s paper in JETP 1971. Bottom line: the formation of a soliton gas in the box problem at large evolution times is indeed predicted by the semi-classical theory but is it what is actually observed in the experiment? If yes, then the authors’ claim should be supported by some quantitative arguments.

SMALLER ISSUES

1. There are some incomprehensible statements in the manuscript like: “DSWs regularize catastrophic discontinuities by mean of rapidly oscillating undular bores” (p.1, second paragraph), or “single phase DSWs are able to generate undular bores” (p. 2, 1st paragraph). In fact, “DSW” and “undular bore” are two different names for the same phenomenon, the latter one being more often used in the fluids context.

2. A very minor, “cosmetic” comment. In the phase diagram in Fig. 1, the sketch of a rogue wave looks more like a fundamental soliton. The presence of two (at least two) side lobes is a “signature” of a rogue wave.

Overall, I believe, the manuscript requires a major revision along the above lines, to be considered for publication in Nature Communications.

Reviewer #2:

Remarks to the Author:

The authors of this work used the connection between topology and the optical beam propagation in nonlinear regime for classification of processes in the beam evolution. This is an interesting approach in nonlinear optics which deserves to be presented to community. The work is reasonably well written, the approach is clear and well illustrated. The paper can be published after the authors will take into account a few comments that are given below.

1. The very first sentence

'In 1967 Gardner, Greene, Kruskal, and Miura developed a mathematical method - the inverse scattering transform (IST) [1] - disclosing the inner features of extreme nonlinear waves in hydrodynamics, plasma physics, nonlinear optics and many other physical systems [2-4].'

is confusing.

Gardner, Greene, Kruskal, and Miura developed the mathematical technique for solving the initial value problems for KdV equation. There is nothing extreme in their work. The word 'extreme' in this sentence is misleading and should be omitted.

2. The following sentence is confusing:

'However, the latter equation is solvable by IST only when the number of degrees of freedom in the IST description is limited.'

It is well known that the NLSE describes the system with infinite number of degrees of freedom. Equivalently, IST description also has an infinite number of degrees of freedom. The number of eigenvalues including those responsible for dispersive radiation is always infinite.

This point should be clarified.

3. The difference between 'small dispersion' NLSE (1) and the ordinary NLSE is only in rescaling the variables ζ and ξ . The parameter ϵ can be eliminated completely from Eq.(1). The only consequence would be then using the 'box' (2) of larger size and longer evolution scale. There is no any point in stressing the use of SDNLSE instead of the normal NLSE. The equivalence of the two approaches should be explained in order to avoid confusing the potential reader.

4. Reference [31] schrödinger => Schrödinger

Reviewer #3:

Remarks to the Author:

Report on the Manuscript#: NCOMMS-19-06027-T

"Topological Control of Extreme Waves" by G. Marcucci, et al.

This interesting paper is focused on new nonlinear theories which state that transitions between extreme waves are allowed [31], thanks to the one-to-one correspondence between the number of wave packet oscillating phases and the genus of toroidal surfaces associated with the nonlinear Schroedinger equation solutions by the Riemann theta function.

The Authors claim to be the first ones to experimentally observe controlled transitions between extreme waves with different genera, varying from dispersive shock waves to Akhmediev Breathers, Peregrine soliton and Soliton gas.

They use a parametric time-dependent nonlinearity to shape the asymptotic wave genus. They consider the box problem in a focusing Kerr-like photorefractive medium and tailor time-dependent propagation coefficients, to explore all the dynamic phases in the nonlinear wave propagation.

The paper contains original results, it is clear and very well written, and may be of interest to different scientific communities.

Therefore I am in principle favorable to its publication in Nature Communications.

Nevertheless, a few points need a clarification in order to definitely validate and/or strengthen the conclusions.

The Authors should address the following major points:

1) The manuscript is based on a rather mathematical paper [31] centered on the one-dimensional focusing nonlinear Schroedinger equation (SDNLSE=small dispersion nonlinear Schroedinger equation, corresponding to eq. (1) in the submitted manuscript).

Here the propagation (longitudinal) coordinate, appearing in the first derivative, is called ζ , while the transverse coordinate, appearing in the second derivative, is called ξ . A coefficient called ϵ appears in the equation in front of the two derivatives (as a linear and quadratic term). In the original paper [31], this coefficient is "a free parameter defining the modulation scale", but what is its physical meaning?

The Authors call ϵ "dispersion parameter". Mathematically, this may be equivalent, but considering the physical problem presented in the paper here I would rather call it diffraction parameter. Am I right or did I miss something?

I think this point is really crucial and needs clarification.

2) In what sense the reported solutions of the NLSE are extreme or rogue waves?

As reported in literature (see for example the same ref [31] here):

"Rogue waves represent the waves of unusually large amplitude $|\psi|_m$, whose appearance statistics deviates from the Gaussian distribution. The conventional "rogue wave criterion" is $|\psi|_m / |\psi|_s > 2$, where $|\psi|_s$ is the significant wave height computed as the average wave height of the largest 1/3 of waves. For the random wave field with Gaussian statistics one has $|\psi|_s = 2 |\psi|_0$, where $|\psi|_0$ is the background (mean field) amplitude, leading to the criterion $|\psi|_m^2 > 8 |\psi|_0^2$."

Would it be possible for the Authors to make a statistical analysis to obtain the PDF of the intensity values (both experimental and numerical), and check which among these waves may be classified

as extreme, by the usual methods? This part could be added as supplementary material in a separated file.

3) I have a fundamental problem in my understanding of the paper. Spatial solitons are waves that remain localized in the transverse coordinate during propagation, thanks to a balance between nonlinearity and diffraction. Here nonlinearity is changing during propagation, due to time dependent photorefractive effect. The present paper is exactly based on this nonlinearity change, responsible of the transitions between the different solution.

Looking at the experimental and numerical Figures, without the mathematical frame given by ref. [31], I see an almost plane wave undergoing a phenomenon of modulational instability/filamentation/pattern formation.

May the Authors clarify in what sense the reported solutions are solitons (Soliton Gas, or Akhmediev Breathers or Peregrine Solitons)? Do these waves propagate for sufficiently long distances in the crystal, compared to the diffraction length?

I apologize if I missed this point in the paper.

4) Finally, a minor point. In the case of small transverse beam, the experimental Peregrine-soliton-like structure profiles shown in Fig. 5c and e present a difference with respect to the analytical PS fitting profile: namely the background value is zero in the experiment, while it is finite in the theoretical fit. May the Authors explain this difference, and why it does not appear in the case of large transverse beam?

In conclusion, the present manuscript may be accepted for publication in nature Communications after clarification of the points stated above.

RESPONSE TO THE REFEREES' REPORTS

Response to Reviewer #1

Reviewer #1: *Referee report for the manuscript "Topological Control of Extreme Waves" by G. Marcucci et al.*

The manuscript reports experimental results on the evolution of an optical pulse with initial near-rectangular intensity profile in a photorefractive crystal with Kerr nonlinearity. The results are interpreted using the recent theoretical developments of Ref [31] where the "box" problem for the small-dispersion focusing nonlinear Schrödinger (NLS) equation was studied analytically and numerically. The main claim of the manuscript is that, by varying the medium's nonlinearity the authors can control the "topology" of the nonlinear coherent structures generated in the optical pulse evolution. The topology is understood in terms of the genus of the underlying spectral Riemann surface and of the associated algebraic theta functions describing the asymptotic NLS solutions in the semi-classical limit. The main idea of the paper looks attractive and promising, and the results are potentially significant. However, there are a number of serious issues, which should be properly addressed before the publication of the manuscript can be considered.

Authors: We thank the Referee for considering our manuscript attractive and promising. We report in the following our detailed answers to the issues that the Referee kindly highlighted and the list of significant changes we made to address all of them. We are trustful that our new results will convince the Referee to suggest acceptance of this deeply revised version of our work.

Reviewer #1: MAIN ISSUES

1. The genus. How can the authors guarantee that the structures they observe have certain genus? In other words, how much of the actual control over the wave "topology" do they have? The fact that the intensity profile they observe can be reasonably well fitted by the shape of a certain NLS breather solution (like in Fig 3 (Akhmediev) or Fig. 5 (Peregrine)) does not necessarily imply that the observed coherent structure is described by this particular NLS solution as the amplitude profile is only a "half" of the complex wave field. To be able to make the identification like this, the phase comparison is necessary along with the amplitude/intensity comparison. Can the authors compare the phase of the experimentally observed structures with that of the theoretical solution to support their claim?

Authors: We thank the Referee for this question, which gave us the possibility to deepen our work and make it more complete. To the best of our knowledge, direct phase measurements of a laser beam propagating in a third-order optical crystal, that is, ruled by a spatial NLSE, have never been performed. There exist examples of phase measurements in Kerr media in fibers, therefore under a temporal NLSE, as reported by A. Mussot *et al.*, Nat. Photon. 12, 303 (2018) and by G. Xu *et al.*, Phys. Rev. E 99, 012207 (2019) (in the latter, phase measurements in a spatial NLSE evolution are shown, but in hydrodynamics). To answer the Referee question, we analyzed phase evolution in the small waist regime, that is, in the Peregrine-like soliton generation, inspired by G. Xu *et al.* paper, cited above. Through interference between the crystal output and a constant signal, we measured the phase shift along the propagation and the transverse direction. These new experimental results profoundly improve our paper. We proved experimentally that the observed light propagation is in remarkable agreement with the one predicted by theory, and in particular that our topological control

allows supervising the transition between genus-zero to genus-two, and therefore to design the resulting genus outcome by tailoring the time of detection. To illustrate these our new experimental observation, we change Fig. 5, removing results at $W_0=50\mu\text{m}$ and adding 4 new panels showing phase evolution in the transition from $g=0$ to $g=2$. Numerical simulations are reported as well.

Action Taken: Page 3, line 32, LHS → We radically changed the third paragraph, regarding the description of Fig. 5, in order to address the modification to the related figure and to describe our experimental and numerical results on phase evolution.

Fig. 5 → We removed intensity outline at $W_0=50\mu\text{m}$ and added 4 new panels showing phase evolution in the transition from $g=0$ to $g=2$, that is, in the Peregrine-like soliton generation, the first three experimentally, while the fourth numerically.

Reviewer #1: *2. Modulational instability. This issue is related to the first one. It is known that the breather structures naturally arise due to the development of modulational instability of a plane wave perturbed by a small noise (D. Agafontsev and V. Zakharov, Nonlinearity, 28 (2015) 2791). Thus, the coherent structures observed in the experiment reported in the manuscript could in principle arise as a result of the development of the noise induced modulational instability in the central part of the box. If the authors insist that the dynamics they observe are dominated by the “genuine” NLS box evolution, they should discuss the influence of the noise, which is inevitable in a physical experiment, and present the corresponding estimates supporting their claims. This issue is crucial, since if the effects the authors observe are dominated by modulational instability, there is no question of any “topological control”.*

Authors: We understand the point of the Referee, and took the occasion to add further analysis to our work, reported in Supplementary Information. We performed experiments and numerical simulations to prove that our results are genuinely caused by a NLSE box evolution, not by MI arising from noise in the central part of the box. Suppl. Fig. 1 reports the experimental outcomes, while simulations are in Suppl. Figs. 1a-e.

MI generates transversal periodic waves, whereas DSWs occur in strongly nonlinear regimes and present fast non-periodic oscillation. From our further analysis, it turns out that MI affects light propagation only for very large beam waists, much larger than the ones analyzed in the main manuscript. This is detailed in Suppl. Fig. 1 and discussed in the Supplementary Information: the MI has a characteristic period much longer than the DSW and occurs in the central part of the box, in correspondance of the flat beam profile. Our new experiments and theoretical analysis clearly enables to discriminate MI dynamics and the box evolution.

Action Taken: Supplementary Information → Section “POTENTIALLY COMPETING EFFECTS: MODULATION INSTABILITY AND LOSSES”, Suppl. Figs. 1,2a-e.

References → We added Ref. [27].

Reviewer #1: *3. Dissipation. Similar to modulational instability, inevitable effects of dissipation should be discussed and estimated. In fact, the dissipation can significantly affect the “local” genus of the breather structure, see S. Randoux et al. Phys. Rev. E 98 (2018) 022219.*

Authors: We are glad that the Referee pointed out such a possible problem, and discussed this issue in Supplementary Information. Since the KLTN crystal is transparent above $\lambda=380\text{nm}$, and the copper doping introduces a small absorption from $\lambda=550\text{nm}$ to $\lambda=800\text{nm}$, our 532nm-laser beam propagation

is not affected by dissipation. Specifically, the linear absorption coefficient is $\alpha=2\text{cm}^{-1}$ for our sample, negligible compared to the propagation length of our experiments (2.5mm).

Due to the relevance of S. Randoux *et al.* paper, we also added Ref. [47] to our manuscript.

Action Taken: Supplementary Information → Section “POTENTIALLY COMPETING EFFECTS: MODULATION INSTABILITY AND LOSSES”

References → We added Ref. [47].

Reviewer #1: 4. DSW evolution and the “topology” control. The wave topology control in the manuscript is verified against the simple formula (5) describing the motion of DSW boundaries (the so-called “breaking curves”) separating the regions with genus 0 and 1, with the point of intersection being associated with the formation of a rogue wave (genus 2). For the comparison of the theoretical velocities (6) with experiment the authors refer to Fig 3.b) showing perfect agreement but I don’t think they explain how the speeds are measured in the experiment. This is important as the breaking curves are not seen by eye on the experimental space-time diagram in Fig 3c. Furthermore, the result presented in Fig. 5 looks very rough. I do not see how it can be compared with Fig. 4 to support the topology control claim.

Authors: We thank the Referee for such a comment and apologize for having been unprecise. For a fixed instant t_1 , the theoretical shock velocity v expressed in Eq. (6) is proportional to the width $\Delta x=W_0*t_1/t_0$ of the plateau between the two DSW in Figs. 2a,3c. Indeed, $\Delta x=2*t_1*v$. We measured $\Delta x/2$ at $t_1=(30\pm 2)\text{s}$ varying the initial power P , so we found $v_1=v/v_0$, with $v_0=L/t_1$ and $L=2.5\text{mm}$ the crystal length. We added this information in the main text, in the labels and the caption of Fig. 3b. About the comparison between Fig. 4a and Fig. 5b (now changed in Fig. 5a), we agree that the former Fig. 4a did not represent appropriately the output detected at small waist regime, whereas we found an excellent agreement with numerical results at $W_0=10\mu\text{m}$. Now we can see in both graphics the absence of shock, and the generation of two Peregrine solitons.

Action Taken: Page 3, line 14, LHS → We modified the first paragraph (the one describing Fig. 3) by adding a final long sentence, in order to explain the experimental shock velocity measurements.

Page 3, line 20, LHS → We modified the second paragraph (the one describing Fig. 3), to take into account the changes of Fig. 4.

Fig. 3 → We change label and caption of Fig. 3b, to take into account the normalized of the experimental shock velocity.

Fig. 4 → We changed Fig. 4a in the intensity outline at $W_0=10\mu\text{m}$ (before was at $W_0=40\mu\text{m}$), in a much better agreement with experimental results.

Reviewer #1: 5. Finite genus solutions vs. rogue waves vs. soliton gas. The connection between rogue waves and finite gap NLS solutions has been discussed in detail in: M. Bertola, G. El and A. Tovbis, *Proc. Roy. Soc.*, 472 (2016) 20160340. In particular, the above authors show that rogue waves can appear, under certain conditions, in the NLS solution of ANY genus greater than one, not necessarily genus 2 (so the association of rogue waves with genus 2 in the phase diagram Fig 1b is misleading). However, the present manuscript implies that any solution with $g>2$ is not a rogue wave but a “soliton gas” (see Fig. 2). This statement is, again, misleading and, strictly speaking, incorrect. In any case, since the authors never explain what they mean by soliton gas it is difficult to meaningfully discuss this issue. At least short discussion should be present in the manuscript, in particular, because the concept of

soliton gas is much less developed than that of a DSW or a rogue wave with just a handful of references following the seminal Zakharov's paper in JETP 1971. Bottom line: the formation of a soliton gas in the box problem at large evolution times is indeed predicted by the semi-classical theory but is it what is actually observed in the experiment? If yes, then the authors' claim should be supported by some quantitative arguments.

Authors: We thank the Referee for such interesting questions, the first one concerning the relation between genera and RWs, the second one about the SG emergence.

Regarding the first issue, we agree with the Referee: in giving at-a-glance picture of the topological signatures of the generated RWs we lacked precision. In order to fix it, we changed Fig. 1b, leaving $g=2$ when referred to the PS (we also changed the RW profile with the analytical PS outline, so the correspondence now is exact), but wrote $g \gg 2$ for SG, and in the caption we changed $g=2$ with $g \sim 2$. Also in the introductory part of the paper, we changed $g=2$ in $g \sim 2$ when defining RWs, and changed $g > 2$ in $g \gg 2$ every time we talk about SGs.

Concerning the SG definition, corrected in $g \gg 2$, we followed the treatise in [40]. G. A. El *et al.* associate the long-time asymptotic solution ψ with a "breather gas" and numerically observe the presence of higher-order RWs with maximum height $4 < |\psi_M| < 5$ in the regions with $g \geq 4$. Their numerical simulations suggest that the pattern of the $\xi - \zeta$ plane (splitting into the regions of different genera) persists as ζ increases, and therefore $g \sim \zeta$ asymptotically. This leads to a disordered finite-density soliton ensemble rather than a well-ordered modulated soliton lattice. In the KdV theory, the thermodynamic type infinite-genus limit of finite-band potentials leads to the kinetic description of a SG [36], but such a treatise is still missing for the NLSE box problem, even in [40]. In our experiments, we see the disordered finite-density soliton ensemble, but the long-time evolution does not allow further analysis, because for $t \gg \tau$ the nonlinearity saturates, hence we lose integrability. These considerations are reported in Supplementary Information.

Action Taken: Page 1, line 2, RHS → We changed $g=2$ with $g \sim 2$.

Page 1, line 2, RHS → We changed $g > 2$ in $g \gg 2$.

Fig. 1B → We changed $g > 2$ in $g \gg 2$ for SG, and in the caption we changed $g=2$ with $g \sim 2$ for RWs.

Supplementary Information → Section "ROGUE WAVES AND SOLITON GAS ANALYSIS IN THE BOX PROBLEM"

References → We added Refs. [35, 44].

Reviewer #1: SMALLER ISSUES

1. *There are some incomprehensible statements in the manuscript like: "DSWs regularize catastrophic discontinuities by mean of rapidly oscillating undular bores" (p.1, second paragraph), or "single phase DSWs are able to generate undular bores" (p. 2, 1st paragraph). In fact, "DSW" and "undular bore" are two different names for the same phenomenon, the latter one being more often used in the fluids context.*

Authors: We see the Referee point, that is, there is no distinction between the definition of DSWs and undular bores, so we changed the sentences reporting both the definitions together.

Action Taken: Page 1, line 13, LHS → We changed "DSWs regularize catastrophic discontinuities by mean of rapidly oscillating undular bores" in "DSWs regularize catastrophic discontinuities by mean of rapid oscillations".

Page 2, line 13, LHS → We changed "These are single-phase DSWs ($g = 1$), able to generate undular bores" in "These wave trains are single-phase DSWs ($g = 1$)".

Reviewer #1: 2. A very minor, “cosmetic” comment. In the phase diagram in Fig. 1, the sketch of a rogue wave looks more like a fundamental soliton. The presence of two (at least two) side lobes is a “signature” of a rogue wave.

Authors: We appreciate the Referee highlighting, indeed we changed the RW profile in a PS analytical outline.

Action Taken: Fig. 1 → We changed the RW profile in Fig. 1b in a PS analytical outline.

Reviewer #1: Overall, I believe, the manuscript requires a major revision along the above lines, to be considered for publication in Nature Communications.

Authors: We sincerely thank the Referee for the highlighted issues. Trustful that our revision is relevant, deep and takes into account all the Referee constructive criticisms and suggestions, we hope the Referee will give her/his approval for the publication of our paper.

Response to Reviewer #2

Reviewer #2: The authors of this work used the connection between topology and the optical beam propagation in nonlinear regime for classification of processes in the beam evolution. This is an interesting approach in nonlinear optics which deserves to be presented to community.

The work is reasonably well written, the approach is clear and well illustrated.

The paper can be published after the authors will take into account a few comments that are given below.

Authors: We thank the Referee for this report. We are glad that the Referee considers our manuscript interesting and worthy of being presented to the community. We took into account every concerning and suggestion, and are hopeful to have clarified all of them in our revised version of the paper.

Reviewer #2: 1. The very first sentence ‘In 1967 Gardner, Greene, Kruskal, and Miura developed a mathematical method - the inverse scattering transform (IST) [1] - disclosing the inner features of extreme nonlinear waves in hydrodynamics, plasma physics, nonlinear optics and many other physical systems [2-4].’ is confusing. Gardner, Greene, Kruskal, and Miura developed the mathematical technique for solving the initial value problems for KdV equation. There is nothing extreme in their work. The word ‘extreme’ in this sentence is misleading and should be omitted.

Authors: We thank the Referee for this clarification about the Gardner *et al.* work. We agree with her/him, for this reason, we omitted the word “extreme” in the first paragraph.

Action Taken: Page 1, line 4, LHS → We removed the word “extreme” in the first paragraph.

Reviewer #2: 2. The following sentence is confusing: ‘However, the latter equation is solvable by IST only when the number of degrees of freedom in the IST description is limited.’ It is well known that the NLSE describes the system with infinite number of degrees of freedom. Equivalently, IST description also has an infinite number of degrees of freedom. The number of eigenvalues including those responsible for dispersive radiation is always infinite. This point should be clarified.

Authors: We agree with the Referee: the reported sentence is confusing and incorrect. To the best of our knowledge, there is no literature about the NLSE box problem solved by IST, even if, in principle,

IST should be able to solve this kind of Cauchy systems, but it becomes unfeasible. On the other hand, the literature about the NLSE box problem solved by finite-gap theory in semiclassical approximation is rich and vast [e.g., R. Jenkins *et al.*, Commun. Pur. Appl. Math. 67, 246 (2014), or G. A. El *et al.*, Nonlinearity 29, 2798 (2016)]. We decided to omit the sentence highlighted by the Referee, and not to enter in problem details, since they are not strictly related to the main purposes of the manuscript.

Action Taken: Page 1, line 23, LHS → We change the sentence “However, the latter equation is solvable by IST only when the number of degrees of freedom in the IST description is limited. Since this number grows as the inverse of the dispersion, when one considers the small-dispersion NLSE (SDNLSE), IST becomes unfeasible” into “However, for the box problem in the small-dispersion NLSE, IST becomes unfeasible.”

Reviewer #2: 3. *The difference between 'small dispersion' NLSE (1) and the ordinary NLSE is only in rescaling the variables ζ and ξ . The parameter ϵ can be eliminated completely from Eq. (1). The only consequence would be then using the 'box' (2) of larger size and longer evolution scale. There is no any point in stressing the use of SDNLSE instead of the normal NLSE. The equivalence of the two approaches should be explained in order to avoid confusing the potential reader.*

Authors: We comprehended the Referee point: the small-dispersion regime is due to a rescaling, directly connected to the size of the box. One could have the same dynamics for $\epsilon=1$ and much larger beam waist. In agreement with the Referee comment, we removed the attribute SD from the definition of the NLSE.

Action Taken: Page 1, line 7 from below, LHS → We removed the acronym SDNLSE.

Page 1, line 15, RHS → We changed SDNLSE in NLSE.

Page 1, line 2 from below, RHS → We changed SDNLSE in NLSE.

Page 2, line 8, LHS → We changed SDNLSE in NLSE.

Page 2, line 26, LHS → We changed SDNLSE in NLSE.

Page 2, line 7 from below, RHS → We changed SDNLSE in NLSE.

Reviewer #2: 4. *Reference [31] schrödinger => Schrödinger*

Authors: We thank the Referee for having highlighted this typo.

Action Taken: References → We corrected Ref. [31] (now Ref. [40]).

Response to Reviewer #3

Reviewer #3: *Report on the Manuscript#: NCOMMS-19-06027-T*

“Topological Control of Extreme Waves” by G. Marcucci, et al.

This interesting paper is focused on new nonlinear theories which state that transitions between extreme waves are allowed [31], thanks to the one-to-one correspondence between the number of wave packet oscillating phases and the genus of toroidal surfaces associated with the nonlinear Schroedinger equation solutions by the Riemann theta function.

The Authors claim to be the first ones to experimentally observe controlled transitions between extreme waves with different genera, varying from dispersive shock waves to Akhmediev Breathers, Peregrine soliton and Soliton gas.

They use a parametric time-dependent nonlinearity to shape the asymptotic wave genus. They consider the box problem in a focusing Kerr-like photorefractive medium and tailor time-dependent propagation coefficients, to explore all the dynamic phases in the nonlinear wave propagation.

The paper contains original results, it is clear and very well written, and may be of interest to different scientific communities.

Therefore I am in principle favorable to its publication in Nature Communications.

Nevertheless, a few points need a clarification in order to definitely validate and/or strengthen the conclusions.

Authors: We sincerely appreciate the Referee opinion on our work, founding it interesting, original and suitable to be published on Nature Communications after some clarifications. We thank the Referee for this kind consideration. We addressed all the doubts she/he raised about the manuscript and produced a Supplementary Information paper as she/he suggested. We believe that the new version of this can satisfy the Referee expectations.

Reviewer #3: *The Authors should address the following major points:*

1) The manuscript is based on a rather mathematical paper [31] centered on the one-dimensional focusing nonlinear Schroedinger equation (SDNLSE=small dispersion nonlinear Schroedinger equation, corresponding to eq. (1) in the submitted manuscript).

Here the propagation (longitudinal) coordinate, appearing in the first derivative, is called ζ , while the transverse coordinate, appearing in the second derivative, is called ξ . A coefficient called ϵ appears in the equation in front of the two derivatives (as a linear and quadratic term). In the original paper [31], this coefficient is “a free parameter defining the modulation scale”, but what is its physical meaning?

The Authors call ϵ “dispersion parameter”. Mathematically, this may be equivalent, but considering the physical problem presented in the paper here I would rather call it diffraction parameter. Am I right or did I miss something?

I think this point is really crucial and needs clarification.

Authors: We thank the Referee for the attention she/he paid in checking our paper. We agree with the summary made above: our work is based on G. A. El *et al.*, Nonlinearity 29, 2798 (2016), and both the experiments and the theoretical analysis we made are related to spatial light propagation in a crystal, where the parameter ϵ does express the diffraction coefficient. The concept of dispersion, even if mathematically equivalent, is physically related to pulses propagation in fibers, that is, to the temporal case. Nevertheless, G. A. El *et al.* in their paper treat ϵ as the dispersion parameter from the beginning to the very ending of the manuscript, talking about the dam break problem in the *small-dispersion* focusing NLSE. With this reference in our mind, we decided to let the definition of ϵ unmodified, to make our treatise in agreement with the mathematical model.

Reviewer #3: *2) In what sense the reported solutions of the NLSE are extreme or rogue waves?*

As reported in literature (see for example the same ref [31] here): “Rogue waves represent the waves of unusually large amplitude ψ_m , whose appearance statistics deviates from the Gaussian distribution. The conventional “rogue wave criterion” is $\psi_m / \psi_s > 2$, where ψ_s is the significant wave height computed as the average wave height of the largest 1/3 of waves. For the random wave field with Gaussian statistics one has $\psi_s = 2 \psi_0$, where ψ_0 is the

background (mean field) amplitude, leading to the criterion $|\psi|_m^2 > 8 |\psi|_0^2$.

Would it be possible for the Authors to make a statistical analysis to obtain the PDF of the intensity values (both experimental and numerical), and check which among these waves may be classified as extreme, by the usual methods? This part could be added as supplementary material in a separated file.

Authors: We are glad that the Referee pointed out this issue. We added Supplementary Information to improve the RW analysis. The reported “RW criterion” is illustrated in SI. The numerical and experimental proofs of this requirement are shown in Figs. 2e-g, 3c, 4e, 5-b. Statistical analysis to obtain the intensity PDF is also exhibited in Suppl. Figs. 2f-i. The latter was studied numerically, in presence and in absence of initial noise, both for small and large waists. Since numerical simulations are in great agreement with experimental results, we refrained from performing further experiments to get other PDFs.

Action Taken: Supplementary Information → Section “ROGUE WAVES AND SOLITON GAS ANALYSIS IN THE BOX PROBLEM”, Suppl. Figs. 2f-i.

Reviewer #3: *3) I have a fundamental problem in my understanding of the paper. Spatial solitons are waves that remain localized in the transverse coordinate during propagation, thanks to a balance between nonlinearity and diffraction. Here nonlinearity is changing during propagation, due to time dependent photorefractive effect. The present paper is exactly based on this nonlinearity change, responsible of the transitions between the different solution. Looking at the experimental and numerical Figures, without the mathematical frame given by ref. [31], I see an almost plane wave undergoing a phenomenon of modulational instability/filamentation/pattern formation.*

May the Authors clarify in what sense the reported solutions are solitons (Soliton Gas, or Akhmediev Breathers or Peregrine Solitons)? Do these waves propagate for sufficiently long distances in the crystal, compared to the diffraction length?

I apologize if I missed this point in the paper.

Authors: We thank the Referee for having given us the occasion to add further analysis to our work, reported in Supplementary Information. We performed experiments and numerical simulations to prove that our results are genuinely caused by a NLSE box evolution, not by MI arising from noise in the central part of the box. Suppl. Fig. 1 reports the experimental outcomes, while simulations are in Suppl. Figs. 1a-e.

MI generates transversal periodic waves, whereas DSWs occur in strongly nonlinear regimes and present fast non-periodic oscillation. From our further analysis, it turns out that MI affects light propagation only for very large beam waists, much larger than the ones analyzed in the main manuscript. This is detailed in Suppl. Fig. 1 and discussed in the Supplementary Information: the MI has a characteristic period much longer than the DSW and occurs in the central part of the box, in correspondance of the flat beam profile. Our new experiments and theoretical analysis clearly enables to discriminate MI dynamics and the box evolution.

Action Taken: Supplementary Information → Section “POTENTIALLY COMPETING EFFECTS: MODULATION INSTABILITY AND LOSSES”, Suppl. Figs. 1,2a-e.

Reviewer #3: *4) Finally, a minor point. In the case of small transverse beam, the experimental Peregrine-soliton-like structure profiles shown in Fig. 5c and e present a difference with respect to the*

analytical PS fitting profile: namely the background value is zero in the experiment, while it is finite in the theoretical fit. May the Authors explain this difference, and why it does not appear in the case of large transverse beam?

Authors: We thank the Referee for this subtle observation. The difference between a theoretical and an experimental PS profile is in the optical intensity/amount of matter/energy source we are considering (depending on the field we are considering, if optics, hydrodynamics or others). If the background is null, it means that the two side lobes are negative, so this cannot represent a positive quantity as the optical intensity. If the background is positive, it means that the energy source of the system is infinite, so this cannot represent an experimental observation. In essence, optical realizations of the PS must be a finite-energy approximation of the analytical PS. In our case, the background is rapidly decaying to zero because, as reported in G. A. El *et al.*, Nonlinearity 29, 2798 (2016), and also in our numerical simulations, the PS describes light behavior only locally. This explains why experimental outcomes and the theoretical fitting in Fig. 5 are in agreement only in the central part, that is, the one of the bump.

In the revised version of our paper, we also report phase measurements showing a further agreement between our results and the analytical PS solution.

Reviewer #3: *In conclusion, the present manuscript may be accepted for publication in nature Communications after clarification of the points stated above.*

Authors: We thank the Referee again and hope that our answers clarified the doubts she/he raised.

List of Changes:

1. Page 1, line 4, LHS → We removed the word “extreme” in the first paragraph.
2. Page 1, line 13, LHS → We changed “DSWs regularize catastrophic discontinuities by mean of rapidly oscillating undular bores” in “DSWs regularize catastrophic discontinuities by mean of rapidly oscillations”.
3. Page 1, line 23, LHS → We change the sentence “However, the latter equation is solvable by IST only when the number of degrees of freedom in the IST description is limited. Since this number grows as the inverse of the dispersion, when one considers the small-dispersion NLSE (SDNLSE), IST becomes unfeasible” into “However, for the box problem in the small-dispersion NLSE, IST becomes unfeasible.”
4. Page 1, line 7 from below, LHS → We removed the acronym SDNLSE.
5. Page 1, line 2, RHS → We changed $g=2$ with $g \sim 2$ and $g>2$ in $g \gg 2$.
6. Page 1, line 15, RHS → We changed SDNLSE in NLSE.
7. Page 1, line 2 from below, RHS → We changed SDNLSE in NLSE.
8. Page 2, line 8, LHS → We changed SDNLSE in NLSE.
9. Page 2, line 13, LHS → We changed “These are single-phase DSWs ($g = 1$), able to generate undular bores” in “These wave trains are single-phase DSWs ($g = 1$)”.
10. Page 2, line 26, LHS → We changed SDNLSE in NLSE.
11. Page 2, line 7 from below, RHS → We changed SDNLSE in NLSE.
12. Page 3, line 14, LHS → We modified the first paragraph (the one describing Fig. 3) by adding a final long sentence, in order to explain the experimental shock velocity measurements.

13. Page 3, line 20, LHS → We modified the second paragraph (the one describing Fig. 3), to take into account the changes of Fig. 4.
14. Page 3, line 32, LHS → We radically changed the third paragraph, regarding the description of Fig. 5, in order to address the modification to the related figure and to describe our experimental and numerical results on phase evolution.
15. Fig. 1 → We changed the RW profile in Fig. 1b in a PS analytical outline. We also changed $g > 2$ in $g \gg 2$ for SG, and in the caption we changed $g = 2$ with $g \sim 2$ for RWs.
16. Fig. 3 → We change label and caption of Fig. 3b, to take into account the normalized of the experimental shock velocity.
17. Fig. 4 → We changed Fig. 4a in the intensity outline at $W_0 = 10 \mu\text{m}$ (before was at $W_0 = 40 \mu\text{m}$), in a much better agreement with experimental results.
18. Fig. 5 → We removed intensity outline at $W_0 = 50 \mu\text{m}$ and added 4 new panels showing phase evolution in the transition from $g = 0$ to $g = 2$, that is, in the Peregrine-like soliton generation, the first three experimentally, while the fourth numerically.
19. References → We corrected Ref. [31] (now Ref. [40]). We added Refs. [17, 27, 30, 33-37, 39, 44, 47, 48]
20. Supplementary Information

Reviewers' Comments:

Reviewer #1:

Remarks to the Author:

The authors have made a very thorough revision of the manuscript following Reviewers' comments. I am happy with the revised version and recommend publication

Reviewer #3:

Remarks to the Author:

The Authors answered to all the points I raised in my first report, therefore, to my opinion the paper is now suitable for publication in Nature Communications